# Learning to think critically about health using digital technology in Ugandan lower secondary schools: A contextual analysis

**Ronald Ssenyonga**[1,2,3], **Nelson K. Sewankambo**[2], **Solomon Kevin Mugagga**[2], **Esther Nakyejwe**[2], **Faith Chesire**[1,4], **Michael Mugisha**[1,5], **Allen Nsangi**[2], **Daniel Semakula**[2], **Matt Oxman**[6,7], **Laetitia Nyirazinyoye**[5], **Simon Lewin**[6], **Margaret Kaseje**[4], **Andrew D. Oxman**[6], **Sarah Rosenbaum**[6]*

1 Faculty of Medicine, Institute of Health and Society, University of Oslo, Oslo, Norway, 2 Department of Medicine, College of Health Sciences, Makerere University, Kampala, Uganda, 3 Department of Epidemiology and Biostatistics, School of Public Health, Makerere University, Kampala, Uganda, 4 Tropical Institute of Community Health and Development, Kisumu, Kenya, 5 School of Public Health, College of Medicine and Health Sciences, University of Rwanda, Kigali, Rwanda, 6 Centre for Epidemic Interventions Research, Norwegian Institute of Public Health, Oslo, Norway, 7 Faculty of Health Sciences, Oslo Metropolitan University, Oslo, Norway

* saro@fhi.no

**Data Availability Statement:** All data files underlying the findings are fully available and without restriction from the Norwegian Centre for

## Abstract

### Introduction

The world is awash with claims about the effects of health interventions. Many of these claims are untrustworthy because the bases are unreliable. Acting on unreliable claims can lead to waste of resources and poor health outcomes. Yet, most people lack the necessary skills to appraise the reliability of health claims. The Informed Health Choices (IHC) project aims to equip young people in Ugandan lower secondary schools with skills to think critically about health claims and to make good health choices by developing and evaluating digital learning resources. To ensure that we create resources that are suitable for use in Uganda's secondary schools and can be scaled up if found effective, we conducted a context analysis. We aimed to better understand opportunities and barriers related to demand for the resources, how the learning content overlaps with existing curriculum and conditions in secondary schools for accessing and using digital resources, in order to inform resource development.

### Methods

We used a mixed methods approach and collected both qualitative and quantitative data. We conducted document analyses, key informant interviews, focus group discussions, school visits, and a telephone survey regarding information communication and technology (ICT). We used a nominal group technique to obtain consensus on the appropriate number and length of IHC lessons that should be planned in a school term. We developed and used a framework from the objectives to code the transcripts and generated summaries of query reports in Atlas.ti version 7.

Research Data database (accession number(s) NSD 2994). The data can be accessed via this web page: http://nsddata.nsd.uib.no/webview/index.jsp?node=0&submode=ddi&study=http%3A%2F%2F129.177.90.161%3A80%2Fobj%2FfStudy%2FNSD2942&mode=documentation&top=yes.

**Funding:** This research was funded by the Research Council of Norway (https://www.forskningsradet.no/en/). Project number 284683, grant no:69006 awarded to ADO. The funders had no role in study design, data collection and analysis, decision to publish, or preparation of the manuscript.

**Competing interests:** The authors have declared that no competing interests exist.

## Findings

Critical thinking is a key competency in the lower secondary school curriculum. However, the curriculum does not explicitly make provision to teach critical thinking about health, despite a need acknowledged by curriculum developers, teachers and students. Exam oriented teaching and a lack of learning resources are additional important barriers to teaching critical thinking about health. School closures and the subsequent introduction of online learning during the COVID-19 pandemic has accelerated teachers' use of digital equipment and learning resources for teaching. Although the government is committed to improving access to ICT in schools and teachers are open to using ICT, access to digital equipment, unreliable power and internet connections remain important hinderances to use of digital learning resources.

## Conclusions

There is a recognized need for learning resources to teach critical thinking about health in Ugandan lower secondary schools. Digital learning resources should be designed to be usable even in schools with limited access and equipment. Teacher training on use of ICT for teaching is needed.

## Background

People everywhere, including youth in Uganda, are exposed to and must deal with health information coming from multiple sources. Health professionals and researchers, charlatans and herbal medicine sellers, governments and international organizations, journalists, advertisers, family members, friends, students and teachers all make claims about the effects of health interventions (e.g. what interventions do and how they work) [1,2]. Many health claims in the public sphere are questionable, and pseudoscience is prevalent–from the information on soft drink bottles, to health-related advertisements that pop up while browsing the internet, to claims about "organic medication" that can cure everything [3]. When people believe unreliable claims, they may end up using ineffective or harmful interventions, which can result in unnecessary suffering and wasted resources [4–8]. Conversely, when people fail to believe and act on *reliable* claims, this can also result in unnecessary suffering and inefficient use of resources [9–11].

A health intervention is any preventive, therapeutic, rehabilitative or palliative action intended to improve the health or wellbeing of individuals or communities [12]. This includes "modern" medicine; lifestyle changes, such as diets or exercise; herbal remedies and other types of "traditional" or "alternative" medicine; public and environmental health interventions; and changes in how healthcare is delivered, financed, or governed. People need to learn to appraise and use information about the effects of health interventions to make better health choices, both at the individual and community level. These skills can be called 'critical health literacy', a part of the broader concept of 'health literacy' which is the ability to access, understand, appraise, and apply health information to make judgments and take decisions in everyday life concerning healthcare, disease prevention, and health promotion [13]. Increased health literacy can improve health and reduce health inequities (5). Improving young people's critical health literacy has the potential to empower them to think critically about health information they are exposed to and to make informed choices as they grow older.

There are several reasons for focusing on improving Uganda's children and young people's ability to think critically about health claims. Uganda has one of the youngest populations in the world [14] with many described as digital natives (born in a digital era) [15]. These young people live in the information age where information is widely accessible online. Although access to the internet can lead to improved access to health information, it also increases peoples' risk of being exposed to and influenced by unsubstantiated claims about health interventions. Additionally, young people have time for learning and can bring knowledge back to their family members and community [16]. They will be the leaders of tomorrow–it is important that they have the skills to make good health decisions and to participate knowledgably in health policy debates.

Previous research has shown that it is possible to teach children as young as 10-years old to assess claims about the effects of health interventions and apply this knowledge in decision making scenarios. The Informed Health Choices (IHC) project developed and evaluated primary school resources that cover 12 concepts from the IHC Key Concept framework for teaching critical thinking about health interventions [17]. A cluster randomized trial with over 10,000 children aged 10–12 years in Uganda showed the resources (printed textbook and exercise book for students; teachers' guide; classroom poster; activity cards for one lesson; two-day teacher training workshop) had a substantial effect [18], and that children retained what they learned for at least one year [19]. In a process evaluation, the IHC researchers found that teachers, parents, and children valued the resources and supported expanding the project to other schools and age groups [20]. However, they also identified two critical barriers to adopting these learning resources: lack of time in school schedules for teaching new content and printing costs.

Few studies have evaluated interventions that teach these key concepts to teenagers [21]. We are therefore developing and evaluating a set of digital learning resources for use by secondary school students (age 14–16), in Uganda. The resources will be digital, to avoid printing costs, and designed to fit into existing school schedules. These stipulations raised the question of availability and use of information and communication technology (ICT) resources in Ugandan secondary schools, as well as how the content might align with the existing curriculum. To explore these and other key issues that could influence the design, uptake, use and implementation of new digital resources in Uganda, we conducted a context analysis before commencing resource development. We conducted similar studies in Rwanda [22] and Kenya.

The objectives of this context analysis were to:

- explore the demand for learning resources used to teach critical thinking about health in Ugandan secondary schools

- map where teaching critical thinking about health best fits in the curriculum

- identify and examine relevant teaching materials that are already in use and, explore conditions for introducing new learning resources

- describe what ICT equipment and software are available for teaching and learning

- identify opportunities and challenges for developing digital learning resources.

## Methods

We used a mixed-methods approach, combining qualitative and quantitative data collection and analysis. We conducted a document analysis to map where critical thinking about health best fits in the curriculum, as well as identify and examine relevant teaching materials.

Through the workshops, we managed to gather consensus on the number of lessons to plan for in a school term as well as duration of these lessons. Data from key informant interviews, focus group discussions and school visits were used to explore the demand for learning resources used to teach critical thinking about health and conditions for introducing new learning resources. We used data from the survey among head teachers and school visits to describe what ICT equipment and software are available for learning and identify opportunities and challenges for developing digital learning resources. For most of the objectives, we used a combination of methods.

## Document analysis

We conducted a document analysis, to map where teaching critical thinking about health best fits in the curriculum, identify and examine relevant teaching materials already in use, and identify opportunities and challenges for developing digital learning resources.

We contacted stakeholders that could help us identify relevant documents. They included teachers in national secondary schools, curriculum specialists at the National Curriculum Development Centre, Commissioners at the Ministry of Education and Sports, and the National Information Technology Authority (NITA) in the Ministry of ICT & National Guidance. Stakeholders identified commonly used textbooks and national curricula documents for nine subjects for the lower secondary school subjects. They also identified digital learning resources for teaching science and health. Through website searches and office visits, we identified policy and planning documents and reports relevant to critical thinking about health and use of digital learning resources. We reviewed the following documents:

- Curricula of nine school subjects (agriculture, chemistry, food and nutrition, general sciences, ICT, mathematics, physical education, physics, and biology

- Two institutional plans (the Sector ICT strategy and Plan for e-Education Vision 2025, and National Information Technology Authority—Uganda Strategic Plan—2018/19–2022/23)

- One policy document in draft (ICT in Education for Uganda dated—Dec 2018)

- Four reports (Integrating ICT into Education in Uganda -July 2014, ICT in Education in Uganda—June 2007, The Contextualized ICT-Competency Framework for Teachers in Uganda—Nov 2016, and the Report of the Technical Audit for the Rural Communications Development Fund (RCDF) to the Uganda Communications Commission Oct 2017)

We conducted a deductive content analysis, using the themes in Table 1. We mapped the IHC Key Concepts [23] against the curricula and textbooks. The IHC Key Concepts are a set of principles for evaluating the trustworthiness of treatment claims, comparisons, and choices, and serve as the basis for developing learning resources. We also examined the extent to which these documents addressed learning to think critically about health and the status of ICT for teaching purposes in Ugandan secondary schools.

Two research assistants independently reviewed each document after a one-day training session.

## Workshops

**Workshop 1—Participants.** We established a network of lower-secondary school teachers to provide input and feedback throughout the project and to participate in an information-gathering workshop. The teachers were from diverse schools, including government and private schools. With the support of the district school inspectors, we selected schools also based on their ICT resources. We included schools that were poorly resourced (schools with no

**Table 1. Codes used to analyze data from document analysis, key informant interviews and focus group discussions.**

| Objective | Themes | Sub-themes |
|---|---|---|
| 1) Explore what demand there is for learning resources for teaching critical thinking about health in secondary schools in Uganda | Teaching critical thinking, health and critical thinking about health | • Critical thinking is taught (Generally)<br>• Critical thinking is not taught (Generally)<br>• How health is taught<br>• Critical thinking about health is taught<br>• Critical thinking about health is not taught<br>• Need for teaching critical thinking about health |
| | Opportunities and challenges for teaching critical thinking about health | • Opportunities to teaching critical thinking<br>• Challenges to teaching critical thinking |
| 2) Map where teaching critical thinking about health best fits in the school curriculum, | Curricula links to critical thinking about health (IHC key concepts) | • Explicitly mentioned IHC concepts<br>• Implicitly mentioned IHC concepts<br>• IHC concepts not mentioned<br>• How critical thinking is taught<br>• Fit in the curriculum<br>• Where to teach critical thinking about health<br>• How much time to teach critical thinking about health |
| 3) Identify and examine relevant resources already in use | Current learning resources used to teach critical thinking, health, and critical thinking about health | • Resources used to teach critical thinking<br>• Resources used to teach health<br>• Resources used to teach critical thinking about health |
| | Current digital learning resources (good and bad examples) | • Good digital learning resources<br>• Poor digital learning resources |
| | Accessibility of the learning resources | • Teachers' access to learning resources<br>• Students' access to learning resources |
| 4) Explore conditions for introducing new learning resources | Decision on learning resources used in schools | • Who decides the learning resources used<br>• Process of decision making about learning resources |
| | Standards for developing digital learning resources | • Existing standards for developing digital learning resources<br>• No standards for developing digital learning resources |
| 5) Describe what ICT facilities and software are likely to be accessible in Ugandan secondary schools for teaching and learning purposes and whether there are any national plans to improve what exists, | ICT facilities available for teaching in secondary schools | • Available ICT conditions for teaching<br>• Planned ICT conditions for teaching<br>• Comparison of ICT facilities<br>• Use of digital facilities for learning. |
| 6) Identify opportunities and challenges for developing digital learning resources. | Opportunities and challenges for using digital learning resources | • Opportunities for use of digital learning resources<br>• Challenges related to use of digital learning resources |

projector and had a student-to-computer ratio of more than 5:1), moderately resourced (schools with no projector and a ratio of 1–3 students to a computer), and well resourced (schools with at least a projector and a ratio of 1 student to 1 computer). We invited 23 teachers via telephone to the workshop and shared with them the objectives of the meeting beforehand, 15 participated.

**Workshop 1—Data collection.** As part of understanding the conditions for introducing new learning resources, we asked teachers about how much time would be practical to allocate to each IHC lesson and what number of lessons could potentially be included in a school term. We adapted the nominal group technique to reach a consensus on the duration of IHC lessons and the number of lessons [24,25]. The steps included:

- Teachers individually generated ideas and submitted free text responses to the questions above via an online survey using SurveyCTO [26].

- RS presented the submitted ideas and responses to the teachers.

- RS moderated a face-to-face session where additional ideas were recorded.

- Teachers held a discussion of the ideas to generate reasons for the suggestions.

- Teachers voted on the ideas that had been refined after the discussion.

We also asked the teachers to provide their views on opportunities and challenges regarding use of digital learning resources in lower secondary schools.

**Workshop 2—Participants and data collection.** We conducted a workshop with curriculum specialists to discuss the same issue of how much time should be realistically planned to teach the IHC learning resources in a school term. Upon receiving administrative clearance, we engaged with a biology specialist who invited three other curriculum specialists for a three-hour face-to-face session. In this session, the three curriculum specialists (for physics, chemistry, and special needs education) reviewed two prototypes of IHC lesson and arrived at a consensus on the number of IHC lessons and duration of the lessons. The prototypes (see Additional file) reviewed were early drafts of lesson plans for the learning resources that we plan to evaluate.

## Key informant interviews and focus group discussions

**Participants.** We conducted face-to-face interviews with individual policy makers, district education officers, head teachers, subject teachers, including some from the teachers' network, students, a developer of digital learning resources, and an online interview with an education officer from a humanitarian aid organization (Table 2).

To select participants from varying contexts, we generated six strata basing on the schools' type of ownership and level of ICT resources. A combination of government and private schools further grouped as poorly-, moderately-, or well-resourced with ICT formed the strata.

**Table 2. Participant in the qualitative interviews.**

| Key informant interviews (number) | Focus group discussions (number of sessions) |
|---|---|
| Ministry of Education and Sports—Commissioner Secondary schools (1) <br> National Curriculum Development Centre—Curriculum Specialist (1) <br> Ministry of Health—Adolescent and School Health (1) <br> District education officers (3) <br> Headteachers (4) <br> Developer of digital learning resources (1) <br> UNICEF (1) | Teachers (1) <br> Students (1) |

We conducted two focus discussion groups, one with teachers and the other with students. We purposively sampled two teachers from each of the six strata. Nine of the twelve teachers invited participated in the focus group discussion that lasted two hours. Using convenience sampling, we recruited a group of students in the focus group discussion from one school, and included males and females from the first, second, and third year of secondary education.

**Data collection.** For all the interviews we used the same semi-structured interview guide (Appendix I) as a starting point and focused on the questions relevant to each informant. As we collected data, we excluded questions that were already covered satisfactorily. We conducted analyses as data were collected, and removed questions for which data saturation was achieved. We modified the guide by adding questions that needed more exploration (e.g., the status of ICT resources in schools considering the COVID-19 pandemic). We used and adapted the same guide for the focus group discussions.

We encouraged participants to speak freely and allowed them to lead the interview in new, relevant directions they thought were important. We conducted face-to-face interviews at the participants' offices, except for one interview held online. RS, who has training in conducting qualitative interviews, led the sessions, while SKM and EN alternated as note takers during the interviews. All participants received information about the study objectives before consenting to be interviewed and audio recorded.

## School visits

We selected 12 schools based on the same six strata regarding ownership and level of ICT resources described above, and visited 10 of these. These schools were different from those the teachers in the focus group discussion came from. During the visits we explored ICT hardware and software availability and use, assessed functionality, and documented how often ICT facilities were used for teaching and learning. We used a prepared checklist (Appendix II) based on questions from the IHC partner responsible for programming the resources, drawing also on relevant items from a set of educational ICT indicators [27]. In addition to completing the checklist for each school, we wrote qualitative descriptions about the state of ICT facilities in schools and students' interaction with ICT during learning. After obtaining written consent from school head teachers, we interviewed them along with teachers, students, and computer lab technicians. We also sought their permission to take photographs and short video recordings.

## School survey

We conducted a telephone survey among a selection of secondary schools in four regions (North, East, West and Central) of Uganda, to learn what ICT facilities were available, their functionality (working conditions), and the extent to which ICT was used for teaching and learning. We contacted participants via phone, due to COVID-19 restrictions that prohibited face-to-face interaction, and because we hoped to achieve a better response rate than with an online survey [28]. We obtained from the Ministry of Education and Sports a list of 3980 secondary schools in Uganda. We selected a sample of 320 schools out of 2746 schools whose telephone contacts were available. This sample size was estimated using the Kish Leslie formula [29] adjusted for multi-level sampling of districts and schools, data from school visits on the availability of projectors as a core ICT facility (eight of the 10 schools), and a 20% non-response rate. We purposively selected the sample from eight strata based on the four regions and two types of school ownership (government and private). We used systematic random sampling to select forty schools from each stratum. Four trained research assistants conducted the telephone survey with both closed and open-ended questions. The questionnaire included

questions regarding ICT equipment and how digital learning resources are, accessed and used (See Additional file).

## Data management and analysis

We collected and entered data from the document review, school visits and telephone survey in Excel and individual and group interviews. For the qualitative data from the telephone survey, we generated descriptive statistics in form of count and percentages using STATA version 14.1. After transcribing and familiarizing ourselves with the qualitative data, two researchers reviewed and independently applied codes developed a priori, as shown in Table 1, to the transcripts in Atlas.ti version 7 [30] and generated query summaries in Microsoft Word and Excel. The researchers harmonized the coding before generating query report summaries of the final version. We reviewed the summary reports and identified key findings. Data saturation was assessed by checking whether similar responses were obtained within and across respondent categories like teachers and school administrators. Two researchers (RS and SN) discussed the key findings and aligned them to the main themes in the framework by study objective. The key findings were shared with four study participants to review and provide feedback before writing them up. We triangulated findings from the document analysis, key informant interviews, focus group discussions, school visits, and telephone survey. We have also included the findings from the consensus workshops on the number and duration of IHC lessons in this report.

For each key finding, we assessed how much confidence to place in the finding using a modified version of GRADE CERQual [31]. The GRADE CERQual assessments provide a systematic and transparent way to make judgments about confidence in findings. Although GRADE CERQual was developed for evidence syntheses rather than for single studies, the components of this approach are suitable for assessing findings from a single study with multiple sources of qualitative data. We adapted the four components:

1. Methodological limitations–the extent to which the data sources underlying a finding are shown to have problems in a way the data were collected;

2. Coherence–assessment of how clear the fit is between the data sources and finding;

3. Adequacy–the degree of richness and quantity of data supporting a finding; and

4. Relevance–the extent to which the finding from the data source is applicable to the context specified in the objective.

Two investigators (RS and SKM) applied the modified GRADE-CERQual approach to each study finding and made a judgment about the overall confidence in the evidence as being high, moderate, low, or very low. We further reviewed and downgraded the overall judgment if there were important concerns for any of the GRADE-CERQual components.

## Ethical considerations

Ethical approval for this study was obtained from the Makerere University School of Medicine Research Ethics Committee and the Uganda National Council of Science and Technology (Reference number HS91ES). A data privacy impact assessment for the project was conducted in accordance with the European Union General Data Protection Regulation and internal approval obtained from the Norwegian Institute of Public Health. We obtained administrative clearance to visit schools from the Ministry of Education and Sports and study district education officers. Before all interviews and observations, we obtained written informed consent from the participants.

## Results

We conducted two workshops with teachers and curriculum developers, analyzed 16 documents, visited 10 schools, and conducted 16 informant interviews, two focus group discussions described in Table 2, and one telephone survey among 238 schools of the 320 reached. To the extent possible and relevant, we triangulated the results across sources. We organized the findings into the following themes:

- Need for teaching critical thinking, health, and critical thinking about health

- Curricula links to critical thinking about health

- Existing learning resources used to teach critical thinking, health, and critical thinking about health

- ICT equipment and software available for learning in secondary schools

- Accessibility and decision-making about learning resources used in schools

- Opportunities and challenges for using digital learning resources

### Need for teaching critical thinking, health, and critical thinking about health

These results are from the qualitative interviews. All the stakeholders interviewed acknowledged a need for teaching critical thinking about health. Many noted the need is pressing, given the rise in the use of untested traditional medicines, number of health intervention available to people to choose from, and access to media platforms like television, radio, and social media, which have exacerbated the spread of unverified information and claims. They also noted that there was an opportunity for equipping young people with critical thinking skills, since they are usually eager to learn and are already making some health-related decisions.

> *"For any human being, critical thinking is key. Before you reach out to getting support from somebody else, you need to be able to interrogate the situation you're in . . . and assess whether you can move forward."* **Teacher in a government-owned school**

> *"The need [for critical thinking] is huge and so crucial. I find it very necessary that children learn such skills because they consume digital content, even parents struggle to talk about health matters with their children."* **Digital content developer**

We identified critical thinking as a core competency in the lower secondary school curriculum. The curriculum specialist, education officers, head teachers, and teachers confirmed that critical thinking is taught across subjects. However, based on the curriculum documents and interviews, we found that critical thinking about health is not explicitly taught in lower secondary schools. Analyzing the subject curricula, we determined that critical thinking about health can be taught as a life skill in most subjects, but perhaps best fits in a science subject like biology. This finding aligns with suggestions from the curriculum specialist and district education officers to incorporate critical thinking about health into existing health related subjects. Tagging the IHC learning resources along existing subjects might ease adoption of learning resources and incorporation of assessments. However, teachers suggested teaching critical thinking about health as a separate subject to give it the emphasis it deserves. They also noted that there are fewer lessons dedicated to health in secondary schools than in primary schools, limiting the number of opportunities for teaching critical thinking about health in existing subjects.

The Uganda National Curriculum Development Center distributes recommended biology and general science textbooks to all national lower secondary schools in both electronic and print versions. There are few learning resources on critical thinking about health. Students and teachers noted that critical thinking about health was mainly taught by calling attention to desirable and undesirable health outcomes, such as encouraging students to make healthy choices by presenting the consequences of specific choices.

*"On the whole, the curriculum is competence based. We want the learners to pick up skills and one of the skills that is clearly highlighted, which we refer to as a generic skill, is critical thinking and problem solving." (**Curriculum specialist**)*

*"In class, they don't tell you go and comb your hair or do this, they only tell you the outcomes like if you don't bathe, you can smell in public" (**Respondent No.9 in the students' focus group discussion.**)*

Despite acknowledging the need for teaching critical thinking about heath, teachers and district education officers noted some challenges. These included:

- Teachers lack teaching materials.

- Teachers are not necessarily critical thinkers themselves.

- It takes time to prepare for lessons, delivery, and assessments.

- Teachers tend to focus the learning on what would be expected on a national examination.

Each school's performance is ranked based on the national examination. Because students and parents prefer highly ranked schools, this compels schools to teach to pass the exam.

*"It can change if we put away the new way of grading and put in the newspapers that one is going to create more time for the children. But as long as we still rank schools depending on the number of first grades, it really limits the time for such activities." (**Headteacher in a privately owned secondary school**)*

## Curricula links to critical thinking about health

These findings were generated from document analysis and the qualitative interviews. The National Curriculum Development Centre finalized a new curriculum for ordinary level (lower) secondary school students in 2019, and rolled it out the following year [32]. The new curriculum focuses on developing higher order thinking skills, which were lacking in the previous curriculum. Teaching has been focused on tests and examinations such as the Primary Leaving Exams. In the new curriculum, critical thinking, problem solving, communication, and ICT proficiency are generic competencies that learners should possess by the end of their school tenure.

Of the 49 concepts in the IHC Key Concepts framework for teaching critical thinking about health interventions [17], only three were explicitly addressed in the new curricula: identifying effects of treatments depends on making comparisons; outcomes should be assessed in the same way in all groups being compared; and people's outcomes should be counted in the group to which they were allocated. The district education officers, health education specialists, and teachers we interviewed confirmed that other IHC Key Concepts were not being taught in the new curriculum.

**Table 3. Example of a learning outcome with suggested activities and assessments from the curricula.**

| TOPIC | COMPETENCY | LEARNING OUTCOMES | SUGGESTED LEARNING ACTIVITIES | SAMPLE ASSESSMENT |
|---|---|---|---|---|
| Sexual Reproduction in Humans | The learner understands that sexual reproduction involves two parents with specialized reproductive systems | a) The learner should be able to:<br>• know the causes, signs and symptoms and understand the mode of transmission of named STIs (Syphilis, Gonorrhoea, Candida, Human Papilloma Virus [HPV], Hepatitis B, and HIV/AIDS).<br>b) appreciate the preventive measures for the named STI's. (Note: The ONLY preventive method recommended for young people is abstinence)<br>c) identify the challenges faced by people living with HIV/AIDS and how to overcome them | As individuals or in groups, learners gather information and report.<br>• Learners listen to a talk from a health worker or watch a video clip about common STIs, and write a report that includes the following:<br> • causes and mode of transmission<br> • signs and symptoms<br> • preventive measures<br>• Learners listen to or recite the song "Alone and Frightened" by Philly Bongoley Lutaya<br>• In groups or as a whole class, learners discuss the stigma/ discrimination portrayed in the song, and the significance of the song in Uganda.<br>Groups write a short play about HIV/ AIDS and attitudes to sufferers. | • Observe pairs and groups engaging in activities. Intervene as necessary and encourage all to participate despite the sensitive nature of some topics.<br>• Listen to learners' discussions and ask questions to encourage creativity and critical thinking.<br>• Evaluate learning as shown by quality of products: oral contributions, annotated diagrams, reports, and posters.<br>• Listen to learners' discussions. Ask probing questions to encourage learners to develop deep understanding of all key issues, and to be sensitive in relation to HIV/ AIDS.<br>Evaluate quality of products: oral contributions, reports, and plays. |

## Existing learning resources used to teach critical thinking, health, and critical thinking about health

The curriculum guides teachers on how content should be taught to students to foster critical evaluation of processes and products that would help them evaluate different health options and reach reasonable decisions. In the curricula [33], there is a dedicated attempt to provide both general and specific illustrations and suggestions of activities that teachers and learners can engage in that can facilitate critical thinking about health, as illustrated in Table 3. These have been incorporated into the textbooks and teacher's guides recommended by National Curriculum Development Centre.

## Accessibility and decision-making about learning resources used in schools

In the documents reviewed and qualitative interviews we found that the government, through the National Curriculum Development Centre, reviews and recommends learning resources. The ministry of education and sports acquires and distributes the learning resources. Some education officers and head teachers noted that introducing new learning resources in schools entails additional work for the teachers. The additional work involved more preparation and delivery time which may take away time for other school activities like sports and drama. Consequently, there is a need to motivate teachers to do this.

*"What happens is that determining content [what students learn] of the students is a role of the National Curriculum Development Centre, now when they have determined the subject content, then this is brought to the school administration and the role of a teacher is to scheme the planning on how it is to be taught and then after that it's the teacher who chooses resources to use for example which textbooks."* **Teacher in a privately-owned school**

*"The good thing with government schools is that there are times when the government acquires books and distributes them to schools and so in that case the school receives what the*

*government recommended, so government provides support in terms of materials to the schools [. . .]. The private schools, the decision is with the head teacher. Whether they are to take on resources recommended by the government or different resources available on the market and it is an open market so to say, so the school has a big say in what they want to buy or what they want to acquire." **Curriculum specialist***

The teachers and curriculum specialists arrived at a consensus that the appropriate number and length of IHC lessons we should plan for was 10 lessons in a school term and 80 minutes each lesson (termed as a double lesson). This amount of time was arrived at by considering the time it would take to teach the example of the lessons we had showed them and for the students to learn this content. We triangulated the finding of 80 minutes with what the National Curriculum Development Centre recommends in the guidelines to schools on implementation of the new lower secondary school curriculum. The National Curriculum Development Centre recommended 40 minutes (termed as a single lesson) to reduce content overload and contact hours in the classroom [34].

### ICT facilities available for teaching in secondary schools

During school visits and telephone survey we obtained data describing ICT facilities and software. Of the 320 head teachers we contacted, 238 (74.4%) responded. The response rate was higher in government schools (86.3%) than private schools (62.5%). The response rates varied from 66.3% to 82.5% across the four regions. See Table 4 for survey results.

Of the 238 schools, 12.6% reported not owning any ICT devices, although some teachers owned personal smartphones or laptops. For schools with ICT devices, most of these were functional at the time of the survey (97.1% of computers and 91.7% of projectors), and they were often used in teaching. Among the schools that owned printers, almost none used them to print learning materials, but instead used them for administrative purposes and in a few cases for printing examinations.

The most common power source in schools is hydroelectric power. Seventy-one percent of the schools surveyed reported access to hydro-electric power, although this varied from 88.7% in the central region to 51.7% in the northern region. Back-up sources like solar and generators, were also reported by the same schools that had reported access to hydro-electric power. Some schools reported cases of postponement of ICT aided lessons when they lost power. Timeliness in fixing faulty ICT equipment used for learning also varied, from within a day to a school term (about three months). Access to digital learning resources was low in schools. In most schools, teachers dictated content (teachers related content orally and students transcribed that); others wrote on the black boards.

In about half (48.7%) of the schools, one or more teachers had delivered at least one lesson using ICT. One in five (23.5%) schools had at least one teacher who had used a smartphone to access and teach content during a lesson. Head teachers reported that the level of confidence among their teachers to deliver lessons using ICT was low (37.8%). Consistent with the Uganda ICT for Education policy, most of the schools indicated that they plan to purchase more ICT equipment like computers and overhead projectors soon.

Most (87.4%) schools provided computer access to students, although this was often limited to class time, ICT lessons, and examinations. Only a few schools (17.6%) allowed access outside class time. Although several government schools had received computers through a government program, most of these had been poorly maintained and became unusable with outdated software and faulty hardware. At the time of our ICT survey and school visits, some of those schools had managed to purchase more computers through contributions from

**Table 4. ICT in School—Survey findings (numbers and percentages).**

| Item | Overall | Ownership | | Region | | | |
|---|---|---|---|---|---|---|---|
| | | *Government* | *Private* | *Central* | *East* | *North* | *West* |
| **Number of schools included** | n = 238 | n = 138 | n = 100 | n = 53 | n = 59 | n = 60 | n = 66 |
| | | | | Percentages | | | |
| **ICT devices available** | | | | | | | |
| *Computers* | | | | | | | |
| Available at school | 87.4 | 89.9 | 84.0 | 94.3 | 81.4 | 86.7 | 87.9 |
| It is functional | 97.1 | 96.0 | 98.8 | 100.0 | 97.9 | 92.3 | 98.3 |
| Used in teaching | 89.6 | 95.0 | 81.9 | 94.0 | 87.2 | 79.2 | 96.5 |
| *Projector* | | | | | | | |
| Available at school | 40.3 | 44.9 | 34.0 | 56.6 | 27.1 | 40.0 | 39.4 |
| Among schools with a projector | | | | | | | |
| It is functional | 91.7 | 93.5 | 88.2 | 96.7 | 93.8 | 87.5 | 100.0 |
| Used in teaching | 92.0 | 94.8 | 86.7 | 93.1 | 100.0 | 76.2 | 50.0 |
| Public address system | 35.3 | 31.9 | 40.0 | 50.9 | 16.9 | 33.3 | 40.9 |
| Electronic boards | 4.2 | 4.3 | 4.0 | 9.4 | 0.0 | 0.0 | 7.6 |
| Television | 67.6 | 69.6 | 65.0 | 81.1 | 57.6 | 56.7 | 75.8 |
| *Power sources* | | | | | | | |
| Electricity (Hydroelectric power) | 71.0 | 72.5 | 69.0 | 88.7 | 64.4 | 51.7 | 80.3 |
| Alternative power source (Generator, solar, etc.) | 35.7 | 36.2 | 35.0 | 54.7 | 18.6 | 31.7 | 39.4 |
| *Material delivery* | | | | | | | |
| Writing on boards (chalk, markers) | 98.3 | 97.1 | 100.0 | 96.3 | 100.0 | 98.3 | 98.5 |
| Using printed notes | 71.8 | 76.1 | 66.0 | 30.1 | 66.1 | 68.3 | 68.2 |
| Dictation (reading to students) of notes during class | 86.1 | 83.3 | 90.0 | 28.8 | 98.3 | 68.3 | 93.9 |
| School central server | 8.8 | 10.9 | 6.0 | 1.3 | 3.4 | 13.3 | 13.6 |
| School website | 2.9 | 2.9 | 3.0 | 2.0 | 0.0 | 1.7 | 4.5 |
| Social media (You tube, WhatsApp, Facebook etc.) | 34.9 | 35.5 | 34.0 | 21.6 | 1.7 | 65.0 | 15.2 |
| Text books | 96.6 | 97.8 | 95.0 | 32.7 | 96.6 | 100.0 | 95.5 |
| Other interactive platforms: Zoom, Skype | 3.4 | 2.9 | 4.0 | 2.0 | 0.0 | 5.0 | 0.0 |
| *Material access* | | | | | | | |
| Textbooks | 81.1 | 75.4 | 89.0 | 33.8 | 100.0 | 96.7 | 93.9 |
| On computers for students to access | 17.6 | 19.6 | 15.0 | 5.2 | 6.8 | 23.3 | 24.2 |
| Local network of computers | 8.8 | 12.3 | 4.0 | 2.6 | 0.0 | 13.3 | 13.6 |
| Social media sites during the COVID-19 pandemic | 20.2 | 21.7 | 18.0 | 19.6 | 1.7 | 10.0 | 16.7 |

parents and school alumni. The ratio of students to computers varied from 1:1 to 50:1, though few schools had achieved a ratio of 1:1. Similarly, ICT resource availability at privately owned secondary schools varied greatly. We observed that student interaction with ICT equipment was rare during the school visits except for scheduled ICT classes, and this was only observed among year four and year six students.

Most of the teachers with whom we interacted during the school visits had delivered a lesson using ICT before the COVID-19 pandemic, but not necessarily at the schools where they were currently teaching. They also mentioned that they did not feel confident delivering lessons using ICT and that it was rare for them to interact with learners in classrooms while using ICT, since most of the content was in handwritten or printed notes and textbooks. The new lower-level secondary school curriculum encourages teachers to use ICT during their interaction with learners. Most of the topics in the curricula contain a guideline caption on

how to use ICT. For example, under the topic of Sexual Reproduction in Humans (Table 3), learners are to listen to or recite the song "Alone and Frightened" via audio [35].

We asked informants to identify good examples of digital learning resources currently used in schools. Most suggested Cyber Schools Technology Solutions [36] and KOLIBRI [37], which they liked because they are interactive and simplify abstract ideas. Cyber Schools Technology Solutions was used to illustrate science concepts and experiments to learners, often simulating a laboratory environment. We also noted that the cost of annual subscriptions was a significant hindrance that prevented many schools from using Cyber Schools Technology Solutions. KOLIBRI is a free, open-source education program. It has modules on life skills and sexual and reproductive health [37].

Most schools used more than one browser, including Mozilla, Google Chrome, Microsoft Edge, Internet Explorer, and Opera. The COVID-19 pandemic had variable effects on schools' ownership and functionality of ICT equipment. Some purchased ICT equipment such as projectors or smart TVs, they fixed faulty computers, and planned to purchase more equipment. For others, their equipment was stolen or became faulty due to poor storage.

## Opportunities and challenges for using digital learning resources

Opportunities and challenges for using digital learning resources are summarized in Table 5. The main opportunity identified from the document analysis and interviews was that there was a committed effort on the part of government to encourage use of ICT in learning, which calls for the development and use of digital learning resources. The curriculum developer mentioned that there is need for teachers and school administrators to develop a mindset shift that 'ICT equipment' can mean other kinds of devices in addition to computers, such as TV, speakers, and smart phones. School closures and the subsequent introduction of online learning during the COVID-19 pandemic has accelerated teachers' use of digital equipment for learning. However, cost is a major barrier. This includes the cost of purchasing ICT equipment, maintenance, software, installing power outlets in classrooms, Internet connections, protection against theft, electricity, back-up power sources, access to digital learning resources, and teacher training.

> *"I think the opportunities are there, people still associate ICT with having computers and once you do that, then you miss out on other opportunities. If schools move away from this mentality and exploit other opportunities like TV, radio, microphones, all those are opportunities, but some people don't look at it that way. But one of the challenges again is resources to acquire those facilities, internet access, electricity in rural areas is a problem, but the opportunities are there. It's just a bit of change of mind set by the school administration, but even a small school with very few resources can afford to have 1 or 2 technologies that they can use."*
> ***Curriculum specialist***

## CERQual summary of the key findings

Our key findings are summarized in Table 6, together with an assessment of confidence in each finding. We assessed all our findings to be of high or moderate confidence.

## Discussion

Given the growing barrage of information on health interventions, compounded by the COVID-19 infodemic [38], it is not surprising that all of the informants we interviewed recognized a need to teach students to think critically about health claims and choices. Moreover,

**Table 5. Opportunities and challenges for using digital learning resources.**

| | Opportunities | Challenges |
|---|---|---|
| **System factors** | | |
| Government level factors | Uganda Communication Commission (under the Ministry of IC and National Guidance) is expanding the internet coverage; therefore, access may improve in the near future | Unreliable power sources with some schools completely lacking access to electricity |
| General factors | • Eases of sharing of and access to learning resources<br>• Improves understanding in the absence of resources to conduct practical experiments.<br>• Has the power to expand students' imaginations and exposure to content that they would have otherwise not come in contact with (films on history of medicine and those on future predictions).<br>• Provide opportunity to look at more resources that may be helpful in generating cohesive understanding of a subject.<br>• Saves time while teaching. Digital resources reduce time it takes to draw illustrations on the boards.<br>• Improves efficiency in conducting research for class engagements<br>• Illustrations in digital resources may also improve the explanations<br>• Although there are few, some schools are ensuring that they hire ICT technician that support teachers during their engagement with learners | High cost of:<br>• Purchasing ICT equipment (computer, projectors, smartphones)<br>• Equipment maintenance (repairs, delicate handling)<br>• Software purchase & upgrade for learning (i.e. Cyber Schools Technology Solutions)<br>• Installing power outlets in classrooms since most do not have these provisions in older buildings.<br>• Internet bundles<br>• Ensuring the classrooms where ICT equipment are stored are theft averse<br>• Using main (hydroelectricity) and alternative (generators, solar, etc.) power/electricity to facilitate learning<br>• Developing digital learning resources (time, skilled personnel)<br>• Some equipment are old with some purchased over 10 years ago and often unusable.<br>• Most schools do not have access to institutional internet to use in teaching and learning; the few that have face limited bandwidth issues.<br>• Compatibility of the software introduced with the technologies with what exists in schools<br>• Shortage of refresher courses to re-school teachers on use of ICT in learning. |
| School level factors | • Solar is also becoming popular although still expensive but this has reduced power outages in some schools<br>• School administrators encourage teachers to participate in trainings that would improve their skills to teach using ICT equipment | • Access to ICT devices to conduct further reading and research is limited, with some schools only allowing access to computer labs during ICT lessons and exams<br>• ICT equipment like projectors, are few in schools and often not fixed in a single room which often takes time takes to set-up and may affect the duration of the lessons<br>• Reduced donations and subsidies in form of ICT equipment when compared to the earlier years of introducing ICT in schools<br>• Theft of ICT equipment like projectors and delays in replenishing<br>• Few technical personnel to support the maintenance of ICT equipment, installation and upgrade of software (anti-viruses, operating systems updates, among others)<br>• Teachers rarely request school administrators to purchase ICT equipment to be used for teaching and learning<br>• Low interest from the school directors to purchase ICT equipment and encourage their teachers to use ICT in teaching and learning |
| **Individual factors** | | |
| Students | The younger learners are usually excited to learn using digital learning resources and often know how to use the ICT | Students may misuse the ICT facilities, for example:<br>• Surfing the internet for pornography<br>• Disrupting class sessions when the phones ring<br>• Contacting friends outside school environment<br>• Addiction to the ICT gadgets |
| Teachers | The new curricular encourages and guides teachers on how to use ICT in their interactions with learners | • Poor attitude towards using ICT in learning<br>• Some teachers are 'digital migrants' and often reluctant to adopt use of ICT in learning<br>• Others feel they are about to exit the teaching system and do not appreciate the need to incorporate ICT in learning<br>• The demand of ICT tools in non-ICT lessons is low<br>• Stuck in the conventional mode of teaching using dictation of notes and writing on boards<br>• Some hold a bias of having students hand write their notes as they dictate content as model that facilitates learning which may not be necessary with digital resources |

**Table 6. CERQual summary of the key findings from the documents analysis, FDGs, Informant interviews and School visits.**

| Theme | Finding | Implication | Sources | Methodological limitations component | Coherence | Adequacy | Relevance | CERQual assessment | Explanation of CERQual assessment |
|---|---|---|---|---|---|---|---|---|---|
| Curricula links to critical thinking about health (IHC Key Concepts) | The new lower secondary school curriculum is aligned with the IHC goal of critical thinking, ICT proficiency, Problem solving, Communication. | The IHC work will be welcomed in the various schools because of the alignment with the curriculum | Curriculum, Interviews | high (most teacher clearly understand these curriculum goals) | high | high | high | high | No concerns regarding methodological limitations, relevance, coherence and adequacy |
| Demand/need for critical thinking about health | There is both a need for teaching critical thinking about health and that for learning resources that foster critical thinking | IHC work is timely and shall fit well in the lower secondary curricula. | FGDs, Key Informants, School visits, Curriculum. | high | high | high | high | high | No concerns regarding methodological limitations, relevance, coherence and adequacy (Students, teachers, a curriculum developer, and education officers all suggested that was a need) |
|  | Most IHC concepts are not taught in the secondary curricula and few places where they are mentioned, they are not explicit. | There is a need to teach these IHC concepts because they are currently not taught | Curriculum & Textbooks, Interviews | high (three sources of data) | high | moderate | high | high | No concerns regarding methodological limitations, relevance, coherence and adequacy |
| Current learning resources used to teach critical thinking, health, and critical thinking about health | There were few learning resources commonly in use across national secondary schools (these included textbooks suggested by the National Curriculum Development Centre shared online and deliver in print at schools) | Resources that facilitate teaching critical thinking, and critical thinking about health, are lacking although perceived as useful. | Textbook & Curriculum, interviews | high | high | moderate | high | high | No concerns regarding methodological limitations, relevance, coherence and adequacy |
| Current ICT conditions for teaching/learning purposes in secondary schools | Some government aided secondary schools and private owned schools have received ICT facilities (like Computers, projectors) from the government and through donations. Almost all have a computer and about less than half have | Despite the existence of ICT facilities in some if not most schools (because ICT is a subject taught at lower secondary level) it is not easy to establish the functionality of these amenities and their use during the student and teacher interaction when learning the non-ICT subjects. However, the new lower secondary school curriculum encourages that ICT is used when learning non-ICT subjects. | Report, Key informants FGDs, School visits | high | high | high | high | high | No concerns regarding methodological limitations, relevance, coherence and adequacy |

(*Continued*)

**Table 6.** (Continued)

| Theme | Finding | Implication | Sources | Methodological limitations component | Coherence | Adequacy | Relevance | CERQual assessment | Explanation of CERQual assessment |
|---|---|---|---|---|---|---|---|---|---|
| Expected ICT conditions for teaching/learning purposes in secondary schools | The policy recognizes the importance of integration of digital technologies in promoting equitable access to e-learning and computer literacy in secondary education. There is a plan to improve ICT facilities for learning as evidenced from the new curricula. Government aided schools are now required to indicate a budget line for ICT when requesting for operational funds. | The IHC digital learning resources when tested in schools with minimal ICT facility should be easily scale-up as more schools continue to install ICT facilities | Policy, ICT in schools survey, Key informants FDGs, | high | high | high | high | high | No concerns regarding methodological limitations, relevance, coherence and adequacy |
| Opportunities for using digital learning resources | • Expectant generation of learners to use ICT in learning. • There is a consented effort by government to lead all secondary schools into procuring digital devices thus improve access and ability to share. | There is a place for using digital learning resources although this may not immediately apply to all secondary schools | Plan, FDGs, Key informants, School visits, ICT in schools survey | high | high | high | high | high | No concerns regarding methodological limitations, relevance, coherence and adequacy |
| Challenges for using digital learning resources | There are still challenges with ICT infrastructure, power/electricity, cost of ICT equipment purchases, maintenance, internet bundles, as well as a shortage of refresher ICT courses for non-ICT teachers | Resources ought to be usable with minimum use of ICT such as a smart phone. Teachers may need to be trained on how to access the digital learning resources. | Report, Policy, FDGs, Key informants, School visits, ICT in schools survey | high | high | high | high | high | No concerns regarding methodological limitations, relevance, coherence and adequacy |

critical thinking skills about health interventions are transferrable to other types of interventions, including agricultural and educational ones [23]. However, a major challenge to teaching critical thinking about health is the absence of appropriate learning resources. Similar to our findings, Nsangi and her colleagues found that despite the feasibility of teaching critical thinking about health in Ugandan primary schools [18], there were no dedicated and appropriate learning resources [17].

We also found that teachers taught critical thinking about health using health outcomes, which focused more on promoting healthy behaviors than critical thinking about what to believe and what to do. In a review of exam script from the old curriculum, only 13% of the questions were higher order thinking questions [39].

Uganda's new lower secondary school curriculum [40] is a competence-based curriculum that emphasizes students' learning of processes and skills rather than teaching that is centered around subject content [41]. This is consistent with a global trend and is similar to curricular changes in other East African countries [42,43]. Critical thinking is a key competence in the Ugandan curriculum, as in most competence-based curricula [44].

As part of mapping where critical thinking best fits in the curriculum, we gathered insights into the plausible duration and number of lessons that could realistically be fitted in a school term. Although teachers reached a consensus that IHC lessons should be 80 minutes (double lessons), the National Curriculum Development Centre recommends that schools limit classroom contact hours to 40 minutes. There is a trade-off between limiting the amount of content that can be taught in 40 versus 80-minute lessons and the challenges of fitting 80-minute lessons into an already overcrowded schedule–in addition to concerns about students' ability to concentrate on a lesson for 80 minutes. After considering this trade-off in the Kenyan, Rwandan and Ugandan contexts, we elected to heed the advice of the Centre to develop learning resources for 40-minute lessons. Even with 80-minute lessons, the amount that can be taught in 10 lessons is limited. Ideally, the IHC Key Concepts should be taught over several school terms and, hopefully, it will be possible to develop additional resources to support this in the future, subject to the findings in the forthcoming intervention trial.

In describing the ICT equipment and software available for use in learning, we identified opportunities and challenges related to use of digital learning resources. ICT can transform education markedly. The global push for incorporating ICT in education has increased access to ICT and reduced the gap in access to educational resources [45]. Over the past 10 years, ICT access has improved in low-income countries [45]. The school closures during the COVID-19 pandemic have motivated several teachers to use ICT to reach the learners. However, there are still major impediments to using ICT in education, including unreliable power supply and ICT-related costs, high student to computer ratio and limited ICT skills among teachers on use of ICT in learning [46,47]. This is consistent with our findings regarding the availability and use of ICT in Ugandan secondary schools. For digital learning resources to be widely used in Uganda and in other low-income countries, it must be possible to use them in a variety of circumstances, not just in schools that are well-equipped with computers. This includes circumstances with frequent power outages, without a stable Internet connection, and without equipment like projectors and computers. Such ICT conditions necessitate development of resources that can be downloaded once on an off line platform accessed on smaller devices like smartphones that require limited upgrades.

There are both similarities and differences in the findings of this context analysis compared to similar studies conducted in Kenya and Rwanda. All three countries have recently or will soon implement new competence-based curricula, but they are at different stages in this process. Rwanda has come furthest in implementing its new curriculum [22,42]. Kenya's new curriculum has not yet been implemented and Uganda is in the process of implementing its new curriculum. All three of the new curricula include critical thinking as a key competence that should be taught across subjects. However, teaching critical thinking skills has not been fully implemented in any of the three countries. Health also is taught across subjects in all three countries, but the proposed new curriculum in Kenya includes health education as a subject [43]. With respect to ICT conditions, all three countries have plans to increase the use of digital learning resources. Rwanda has also come further than Kenya and Uganda in this regard, with over half of public schools having at least two computer laboratories with laptops and an Internet connection. There is still limited use of digital learning resources in all three countries.

## Strengths and limitations

The new curriculum was only rolled out a few weeks before schools went into lock down for seven months due to the COVID-19 pandemic. Consequently, there are many uncertainties on how the context might change in the near or mid-term given the changes brought about directly or indirectly by the COVID-19 pandemic. These changes may affect implementation

of the curriculum in which new digital learning resources will be utilized. We also were not able to observe ICT use by students in lower secondary school (years one to three), due to school closures. We were, however, able to observe year-4 and year-6 students and gathered a sense of student accessibility to school ICT facilities, although this evidence could be termed as partially relevant given our target group of younger adolescents in year-2 and year-3.

The use of multiple qualitative methods and mixed methods improved the validity and certainty of the study findings.

## Conclusions

Teaching critical thinking about health is not deliberate, but there is still opportunity to make it so in the new lower secondary school curriculum, especially given the need expressed by curriculum developers, teachers and students. There are few learning resources to teach critical thinking about health, and these mainly focus on promoting health behaviors rather than thinking critically. Lower national secondary schools vary widely in their access to and use of digital equipment for teaching and learning. Other contextual conditions also vary considerably, such as stability of power connections and teachers' technology skills and attitudes toward use of ICT for teaching. Although improvements in different contextual elements in the Ugandan schools is being planned for by government in the mid- and long-term, the digital learning resources to be developed need to be quite flexible so that they can be used in different ICT contexts, including teachers with little ICT experience.

## Supporting information

**S1 Checklist. ISSM_COREQ_Checklist_form.**
(PDF)

**S1 File. CerQual assessment of key findings.**
(XLSX)

**S2 File. Links to the learning resources.**
(DOCX)

**S3 File. Questionnaire–ICT school survey conducted via telephone.**
(DOCX)

## Acknowledgments

We thank the schools, teachers, members of the teacher's network, and students who participated in this study. We also thank the research assistants who participated in the collection of survey data and review of the analyzed documents. We also appreciate Ms. Stellah Namatovu (SN) who participated in the coding and review of the interview data. We also acknowledge the Ministry of Education and Sports, and the National Curriculum Development Centre that provided administrative letters of support to schools and access to relevant documents.

## Author Contributions

**Conceptualization:** Ronald Ssenyonga, Nelson K. Sewankambo, Faith Chesire, Michael Mugisha, Allen Nsangi, Daniel Semakula, Matt Oxman, Laetitia Nyirazinyoye, Margaret Kaseje, Andrew D. Oxman, Sarah Rosenbaum.

**Data curation:** Ronald Ssenyonga, Solomon Kevin Mugagga, Esther Nakyejwe.

**Formal analysis:** Ronald Ssenyonga, Solomon Kevin Mugagga, Esther Nakyejwe, Simon Lewin, Sarah Rosenbaum.

**Funding acquisition:** Andrew D. Oxman.

**Validation:** Ronald Ssenyonga, Nelson K. Sewankambo, Solomon Kevin Mugagga, Esther Nakyejwe, Faith Chesire, Michael Mugisha, Allen Nsangi, Daniel Semakula, Matt Oxman, Simon Lewin, Margaret Kaseje, Andrew D. Oxman, Sarah Rosenbaum.

**Writing – original draft:** Ronald Ssenyonga.

**Writing – review & editing:** Ronald Ssenyonga, Nelson K. Sewankambo, Solomon Kevin Mugagga, Esther Nakyejwe, Faith Chesire, Michael Mugisha, Allen Nsangi, Daniel Semakula, Matt Oxman, Laetitia Nyirazinyoye, Simon Lewin, Margaret Kaseje, Andrew D. Oxman, Sarah Rosenbaum.

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
