## [Decision Letter · Decision Letter 0]

26 Aug 2021

PONE-D-21-14333

Learning to think critically about health using digital technology in Ugandan lower secondary schools: a contextual analysis.

PLOS ONE

Dear Dr. Rosenbaum,

Thank you for submitting your manuscript to PLOS ONE. After careful consideration, we feel that it has merit but does not fully meet PLOS ONE’s publication criteria as it currently stands. Therefore, we invite you to submit a revised version of the manuscript that addresses the points raised during the review process.

Both reviewers agree on the importance of this study and its relevance. Yet, especially the second reviewer has major important concerns that need to be carefully addressed. The manuscript thus needs extensive revisions.

We look forward to receiving your revised manuscript.

Kind regards,

Sara Rubinelli

Academic Editor

PLOS ONE

Journal Requirements:

Reviewers' comments:

Reviewer's Responses to Questions

**Comments to the Author**

1. Is the manuscript technically sound, and do the data support the conclusions?

Reviewer #1: Yes

Reviewer #2: Partly

2. Has the statistical analysis been performed appropriately and rigorously? 

Reviewer #1: Yes

Reviewer #2: I Don't Know

3. Have the authors made all data underlying the findings in their manuscript fully available?

Reviewer #1: No

Reviewer #2: Yes

4. Is the manuscript presented in an intelligible fashion and written in standard English?

Reviewer #1: Yes

Reviewer #2: Yes

5. Review Comments to the Author

Reviewer #1: This article describes a very interesting study of a project to support the development of critical thinking about health interventions among pupils in selected secondary schools in Uganda. This topic fits well with the transition of the Ugandan school curriculum to a competence-based framework.

I recommend publishing this contribution after minor revisions:

1. A wider international audience may need one sentence of context about the song "Alone and Frightened" and/or a link to information.

2. The description of the availability of electricity isn't sufficiently clear. It seems that overall 71% of schools have mains electricity (that is predominantly hydroelectric power in Uganda) and the remainder have an electricity generator or solar power generated locally.

3. There is insufficient comparison with the results of a recently published research work that has several authors in common and on a very similar topic in reference 22 (Mugisha M, Uwitonze AM, Chesire F, Senyonga R, Oxman M, Nsangi A, et al. Teaching critical thinking about health using digital technology in lower secondary schools in Rwanda: A qualitative context analysis. PLoS One. 2021;16(3).).

Reviewer #2: This paper addresses an extremely interesting topic. Critical thinking skills about health, or Critical Health Literacy, could be taught, but there are few interventions existing and it is necessary to investigate their opportunity and feasibility. The authors have conducted a thorough contextual analysis, but I believe the manuscript in its present form it is suitable for publication and does not do justice to the work. Below my main concerns:

• The authors list a number of research questions in the introduction and describe a number of studies in the methods section. However, it is not clear which study helps answering which research question. I believe this should be specified.

• In the methods section some information is missing throughout.

o Document analysis: When listing the documents, it is not clear how they have been selected and why. In addition, information is missing on the process of analysis (inductive?) and on how table 1 has been derived. It is not clear to me what “two-degree holders” means, it not specific of expertise of the reviewers.

o Workshops: the description of the workshops is not specific about their structure. How were participants informed beforehand? How was NGT implemented? Also, wen mentioning prototypes in workshop 2, more details would be appreciated.

o Qualitative interviews: when explaining data collection the authors indicate, “we excluded questions that were already covered satisfactorily in the preceding interviews and modified the guide by adding questions that needed more exploration”. Could you be more clear about this? Was analysis undergoing in parallel? Do you mean that data saturation was achieved? More information about the analysis is needed as well.

o School survey: could you please provide more information about the questionnaire?

• The results section presents results from the different studies, but is not evident which study contributed to what. It would be good, in my opinion, to clarify this both in the introduction to the results section and in the other subchapters.

• I am getting lost in the second part of the discussion (section 4 and 5) as I do not see the direct link with the purpose of the study and the results presented. Maybe it would be worth it to elaborate this content further.

6. PLOS authors have the option to publish the peer review history of their article (what does this mean?). If published, this will include your full peer review and any attached files.

Reviewer #1: No

Reviewer #2: No

---

## [Author Response · Author response to Decision Letter 0]

4 Oct 2021

Re: Response the comments on our manuscript; PONE-D-21-14333

Learning to think critically about health using digital technology in Ugandan lower secondary schools: a contextual analysis.

PLOS ONE

We wish to thank you for the helpful comments. We have carefully reviewed and addressed each comment as detailed below;

Comments from the academic editor

Response: We have checked the PLOS ONE style requirements from the guidelines above and edited our paper to ensure that we adhere to them. 

2. Please include your tables as part of your main manuscript and remove the individual files. Please note that supplementary tables (should remain/ be uploaded) as separate "supporting information" files

Response: We have included the tables as part our main manuscript and removed the individual files as guided. 

Response: We have corrected this and now the grant information in both sections matches.

Reviewers' comments:

Reviewer #1: 

1. A wider international audience may need one sentence of context about the song "Alone and Frightened" and/or a link to information.

Response: We have added a reference on page 23 (see line 493 in the revised manuscript track changes version). We think that the reference would provide more details about the song than a sentence. 

2. The description of the availability of electricity isn't sufficiently clear. It seems that overall 71% of schools have mains electricity (that is predominantly hydroelectric power in Uganda) and the remainder have an electricity generator or solar power generated locally.

Response: We have re-written this paragraph to clarify that only 71% of the schools had hydroelectric power which is the most common source of power in schools and that 29% did not have access to any form of electricity on page 21 (see line 456-460 in the revised manuscript track changes version). 

3. There is insufficient comparison with the results of a recently published research work that has several authors in common and on a very similar topic in reference 22 (Mugisha M, Uwitonze AM, Chesire F, Senyonga R, Oxman M, Nsangi A, et al. Teaching critical thinking about health using digital technology in lower secondary schools in Rwanda: A qualitative context analysis. PLoS One. 2021;16(3).).

Response: We have improved the comparison by adding a paragraph that enriches the comparison as well as referencing the areas where similar findings were found on page 32 (see line 548 in the revised manuscript track changes version). 

Reviewer #2: 

This paper addresses an extremely interesting topic. Critical thinking skills about health, or Critical Health Literacy, could be taught, but there are few interventions existing and it is necessary to investigate their opportunity and feasibility. The authors have conducted a thorough contextual analysis, but I believe the manuscript in its present form it is suitable for publication and does not do justice to the work. Below my main concerns:

• The authors list a number of research questions in the introduction and describe a number of studies in the methods section. However, it is not clear which study helps answering which research question. I believe this should be specified.

Response: We have edited the first paragraph in the methods section on page 6 (see line 128 and 136 in the revised manuscript track changes version) to clarify which objectives are addressed by the different methods that we used. 

• In the methods section some information is missing throughout.

o Document analysis: When listing the documents, it is not clear how they have been selected and why. In addition, information is missing on the process of analysis (inductive?) and on how table 1 has been derived. It is not clear to me what “two-degree holders” means, it not specific of expertise of the reviewers.

Response: We have added a sentence that justifies why and how we selected the listed documents. Documents relevant to critical thinking about health and ICT in education were selected from the website searches and office visits (see line 148 to 150 on page 7 in the revised manuscript track changes version). We have also specified that we conducted deductive analysis of the documents (see line 161 in the revised manuscript track changes version). We have also rewritten the “two-degree holders” to two research assistants to demonstrate the expertise of the reviewers on page 7 (see line 168 in the revised manuscript track changes version) all on page 7

o Workshops: the description of the workshops is not specific about their structure. How were participants informed beforehand? How was NGT implemented? Also, wen mentioning prototypes in workshop 2, more details would be appreciated.

Response: We have added a sentence about how participants were informed beforehand on page 8 (see line 179 and 181 in the revised manuscript track changes version). The NGT was implemented as indicated in the five bullet points on page 8 (see line 187 to 192 in the revised manuscript track changes version). We have provided more details about the prototypes in workshop 2 for on page 9 (see line 202 to 204 in the revised manuscript track changes version).

o Qualitative interviews: when explaining data collection the authors indicate, “we excluded questions that were already covered satisfactorily in the preceding interviews and modified the guide by adding questions that needed more exploration”. Could you be more clear about this? Was analysis undergoing in parallel? Do you mean that data saturation was achieved? More information about the analysis is needed as well.

Response: We have clarified that the analysis was conducted as data were collected and that questions were removed when saturation was achieved on page 9 (see line 224 to 225 in the revised manuscript track changes version). Data saturation was assessed by checking whether similar responses were obtained within and across respondent categories as detailed in the description of the analysis on page 11 (see line 275 to 277 in the revised manuscript track changes version). 

o School survey: could you please provide more information about the questionnaire?

Response: We have added a sentence at the end of the school survey paragraph under methods that shows the sub-sections that we included in the questionnaire when conducting the school survey among head teachers on page 11 (see line 264 and 265 in the revised manuscript track changes version). 

• The results section presents results from the different studies, but is not evident which study contributed to what. It would be good, in my opinion, to clarify this both in the introduction to the results section and in the other subchapters.

Response: We have addressed this in the first paragraph of the Methods section, as indicated above in response to this reviewer’s first comment. In the results section, we have indicated which methods have contributed to each section: on page 15 (see line 312 to 314 in the revised manuscript track changes version). 

Page 15 line 328, page 17 line 383, page 18 line 410. In addition, the sources for each key finding are specified in Table 6.

• I am getting lost in the second part of the discussion (section 4 and 5) as I do not see the direct link with the purpose of the study and the results presented. Maybe it would be worth it to elaborate this content further.

Response: We have added a sentence at the beginning of each of those paragraphs clarifying how they related to the results on page 31 (see lines 551 to 552 and 564 to 566 and in the revised manuscript track changes version)

---

## [Decision Letter · Decision Letter 1]

9 Nov 2021

Learning to think critically about health using digital technology in Ugandan lower secondary schools: a contextual analysis.

PONE-D-21-14333R1

Dear Dr. Rosenbaum,

We’re pleased to inform you that your manuscript has been judged scientifically suitable for publication and will be formally accepted for publication once it meets all outstanding technical requirements.

Kind regards,

Sara Rubinelli

Academic Editor

PLOS ONE

Additional Editor Comments (optional):

Reviewers' comments:

Reviewer's Responses to Questions

**Comments to the Author**

1. If the authors have adequately addressed your comments raised in a previous round of review and you feel that this manuscript is now acceptable for publication, you may indicate that here to bypass the “Comments to the Author” section, enter your conflict of interest statement in the “Confidential to Editor” section, and submit your "Accept" recommendation.

Reviewer #1: All comments have been addressed

Reviewer #2: All comments have been addressed

2. Is the manuscript technically sound, and do the data support the conclusions?

Reviewer #1: Yes

Reviewer #2: Yes

3. Has the statistical analysis been performed appropriately and rigorously? 

Reviewer #1: Yes

Reviewer #2: Yes

4. Have the authors made all data underlying the findings in their manuscript fully available?

Reviewer #1: Yes

Reviewer #2: Yes

5. Is the manuscript presented in an intelligible fashion and written in standard English?

Reviewer #1: Yes

Reviewer #2: Yes

6. Review Comments to the Author

Reviewer #1: All of my qustions with regard to this manuscript have been addressed by the authors and I recommend publication of this paper.

Reviewer #2: The authors have addressed all comments from the previous review. The manuscript is now more coherent and the relation between objectives, methods and results is more linear.

7. PLOS authors have the option to publish the peer review history of their article (what does this mean?). If published, this will include your full peer review and any attached files.

Reviewer #1: No

Reviewer #2: No

---

## [Editor Report · Acceptance letter]

24 Jan 2022

PONE-D-21-14333R1 

Learning to think critically about health using digital technology in Ugandan lower secondary schools: a contextual analysis. 

Dear Dr. Rosenbaum:

I'm pleased to inform you that your manuscript has been deemed suitable for publication in PLOS ONE. Congratulations! Your manuscript is now with our production department. 

Kind regards, 

on behalf of

Dr. Sara Rubinelli 

Academic Editor

PLOS ONE